# Integrating a Luminescent Porous Aromatic Framework into Indicator Papers for Facile, Rapid, and Selective Detection of Nitro Compounds

**DOI:** 10.3390/molecules27196252

**Published:** 2022-09-22

**Authors:** Bo Cui, Changyuan Gao, Jiating Fan, Jinni Liu, Bin Feng, Xianghui Ruan, Yajie Yang, Ye Yuan, Kuo Chu, Zhuojun Yan, Lixin Xia

**Affiliations:** 1College of Chemistry, Liaoning University, Shenyang 110036, China; 2School of Environmental Science, Liaoning University, Shenyang 110036, China; 3Key Laboratory of Polyoxometalate and Reticular Material Chemistry of Ministry of Education, Faculty of Chemistry, Northeast Normal University, Changchun 130024, China; 4Liaoning Key Laboratory of Chemical Additive Synthesis and Separation, Yingkou Institute of Technology, Yingkou 115014, China

**Keywords:** porous aromatic framework, luminescence, nitro compounds, indicator paper, selective detection

## Abstract

Porous aromatic framework materials with high stability, sensitivity, and selectivity have great potential to provide new sensors for optoelectronic/fluorescent probe devices. In this work, a luminescent porous aromatic framework material (LNU-23) was synthesized via the palladium-catalyzed Suzuki cross-coupling reaction of tetrabromopyrene and 1,2-bisphenyldiborate pinacol ester. The resulting PAF solid exhibited strong fluorescence emission with a quantum yield of 18.31%, showing excellent light and heat stability. Because the lowest unoccupied molecular orbital (LUMO) of LNU-23 was higher than that of the nitro compounds, there was an energy transfer from the excited LNU-23 to the analyte, leading to the selective fluorescence quenching with a limit of detection (LOD) ≈ 1.47 × 10^−5^ M. After integrating the luminescent PAF powder on the paper by a simple dipping method, the indicator papers revealed a fast fluorescence response to gaseous nitrobenzene within 10 s, which shows great potential in outdoor fluorescence detection of nitro compounds.

## 1. Introduction

Nitro compounds are an important chemical raw material in medicine, dyes, spices, explosives, and other industries. However, with the widespread use of nitro compounds, their hazards have gradually led to the wide concern that they will be discharged into the groundwater and soil, and then cause serious pollution to ecosystems through biological pathways [1,2]. For example, nitrobenzene, a non-degradable toxin, could cause nausea, vomiting, pulmonary dysfunction, genotoxicity, and severe brain disease after excessive deposition in the human body. Therefore, a lot of scientific research has been devoted to exploring the trace detection of nitro compounds [3]. The fluorescence detection method has several distinctive characteristics of portability, fast response, and high sensitivity, and has been considered a reliable technology for wide usage in the detection of nitro compounds [4,5,6]. To achieve facile and sensitive fluorescence detection of nitro compounds, one of the research challenges is the synthesis of target substances with high sensitivity and anti-interference capability.

Characterized by pure organic structural components, porous organic polymers (POPs) have attracted wide attention due to their rigid skeleton and large surface area [7,8,9,10,11]. Among them, porous aromatic frameworks (PAFs) are composed of strong carbon–carbon bonds connecting aromatic building units, which makes them possess good stability in harsh application environments. Compared with traditional porous solids of inorganic and inorganic–organic hybrid architectures, the surface and structure of PAF materials can be easily fine-tuned to suit different applications, such as catalysis, adsorption, energy storage, fluorescence detection, etc. [12,13,14,15,16]. Especially in the field of fluorescence detection, the materials have unique optoelectronic properties by constructing large delocalized π-conjugated structures [17,18,19,20]. Zhao et al. improved the sensitivity of porous organic polymer MFCMP-1 to explosive molecules (such as nitrobenzene, 2-nitrotoluene, and 2,4-dinitrotoluene) by exploiting the high-emission and three-dimensional π-conjugated characteristics of the framework [21]. Yuan et al. exploited the σ*-π* conjugation generated by the interaction between the σ* orbital of germanium and the π* orbital of the benzene ring to enhance the sensitivity of electron delocalization [22]. Ma et al. synthesized a core-shell porous aromatic framework that enables the selective detection of nitro compounds TNT and TNP by tuning the LUMO level of core-shell PAFs [23], etc. [24,25,26]. These studies show that porous organic polymers have great application prospects in the fluorescence detection of nitro-explosives. However, the research on this issue is still in its infancy; a simple POP sample containing large delocalized π-conjugated units should be proposed to illustrate the fluorescence detection mechanism of POP networks. 

In this study, a fluorescent porous aromatic framework (PAF), LNU-23, was synthesized through a Suzuki cross-coupling reaction using tetrabromopyrene and 1,2-diphenyldiboronic acid pinacol ester as the building monomers. Due to the introduction of the fluorescence emission pyrene group, electrons quickly migrate along the polymer chain to the acceptor for the amplification of the fluorescence signal, and improve the sensitivity [27]. Accordingly, the PAF powder-dipped paper effectively real-time monitored nitro compounds in their respective solid, gas, and liquid states.

## 2. Results and Discussion

LNU-23 was synthesized through the Suzuki cross-coupling reaction of tetrabromopyrene and 1,2-bisphenyldiboronic acid pinacol ester in the catalytic system of Pd[P(C_6_H_5_)_3_]_4_/K_2_CO_3_/DMF (Figure 1). FT-IR spectra of the PAF polymer and its raw monomers were shown in Figure 2a. Compared with the monomers, the typical peaks assigned to the stretching vibration of C-B bond at 1417 cm^−1^ and B-O bond at 1351 cm^−1^, respectively, disappeared in the spectrum of LNU-23 polymer. Meanwhile, there was no characteristic absorption band at 500 cm^−1^ in the FTIR spectrum of LNU-23 that demonstrated the successful preparation of polymeric network. The ^13^C CP/MAS NMR spectrum further confirmed the structural integrity of LNU-23 (Figure 2b). The chemical shift at 143 ppm belonged to the carbon atoms connected to the pyrene ring or the benzene ring; the main signal at 135 ppm was attributed to the carbon in the pyrene group. There was a broad peak at 127 ppm corresponding to the unsubstituted phenyl and pyrene carbons [28]. Both FT-IR and ^13^C NMR spectra were well consistent with the proposed structure of the LNU-23 polymer.

The powder X-ray diffraction (PXRD) spectrum indicated that LNU-23 possessed an amorphous structure (Appendix A). This is mainly due to the free rotation of phenyl linkers and the uncorrected binding patterns of LNU-23 oligomers result in architectural defects and an irregular whole structure [29]. Thermal stability for LNU-23 was evaluated under a nitrogen atmosphere. As shown in Appendix A, ~8% weight loss occurred at around 400 °C, which was attributed to the decomposing of the LNU-23 skeleton, indicating its excellent thermal stability [30,31]. Scanning electron microscopy (SEM) showed that the PAF sample was composed of irregular particles. Transmission electron microscopy (TEM) illustrated a similar result of the amorphous network of LNU-23 (Figure 2c,d). The porous property of LNU-23 was evaluated by measuring N_2_ adsorption–desorption isotherms at 77 K. As shown in Appendix A, the adsorption–desorption curve of the material exhibits a large hysteresis loop, indicating the existence of mesopores in the LNU-23 network. The pore width was mainly distributed in 5 and 7.5 nm and evaluated by the slit/cylindrical non-local density functional theory (NLDFT) model. The BET-specific surface area of LNU-23 was calculated to be 10 m^2^ g^−1^ in the range of P/P_0_ = 0.03–0.25. This low specific surface area was due to the large π-conjugated segments in the LNU-23 network forming a tightly packed structure [32]. 

As shown in Appendix A, the ultraviolet visible absorption spectra of the monomers and LNU-23 were measured in absolute ethanol. The absorption wavelengths of the monomers were located in the range of 250–400 nm. Compared with the 1,3,6,8-tetrabromopyrene monomer, the polymer LNU-23 showed an obvious red shift. In addition, the luminescence intensity of LNU-23 was significantly increased, which was ascribed to the fact that the stable conjugated structure of LNU-23 reduced the energy consumption during vibrational relaxation, thus enhancing the luminescence intensity [33]. Under 365 nm UV light irradiation, the LNU-23 powder exhibited green fluorescence with a fluorescence quantum yield of 18.31% (Appendix A and Figure 3a). The outstanding fluorescence property of LNU-23 makes it a promising material in the sensing field, which urges us to explore the application of monitoring nitro compounds through fluorescence sensing.

The polymer LNU-23 was dispersed in tetrahydrofuran solvent, and then added dropwise to respective benzene (B), toluene (MB), bromobenzene (BB), monochlorobenzene (MCB), p-nitrophenol (PN), and nitrobenzene (NB), to study the fluorescent quenching capability of LNU-23. As seen in Figure 3a, the fluorescent LNU-23 was not quenched in all non-nitro solvents. On the contrary, it revealed an obvious quenching phenomenon in various nitro compounds of PN and NB. There was an obvious quenching phenomenon that can be observed by the naked eye in Figure 3b. It demonstrated that LNU-23 possessed a real-time visual on-site fluorescence detection of nitro compounds. 

To explore the sensing ability of LNU-23, luminescence quenching experiments were performed with different amounts of nitrobenzene. As seen in Figure 3c, the fluorescence intensity of LNU-23 gradually decreased with the increase in the nitrobenzene amount. The quenching efficiency was calculated according to the formula (*I*_0_ − *I*)/*I*_0_ × 100% (*I*_0_ is the initial fluorescence intensity, *I* is the fluorescence intensity after quenching) [34]. When the concentration of nitrobenzene reached 3.91 × 10^−2^ mol L^−1^, the quenching efficiency of LNU-23 was calculated to be 98.24%, indicating that LNU-23 exhibited a good fluorescence quenching ability for nitro compounds (Figure 3d). The fluorescence quenching efficiency was further analyzed using the Stern–Volmer (SV) equation (Appendix A), *I*_0_/*I* = *K_SV_M*, where *M* is the analyte concentration, and *I*_0_ and *I* are the emission intensities in the absence and presence of the analyte, respectively. The correlation of the quenching effect (*I*_0_/*I*) to the nitrobenzene concentration showed a good linear relationship (R^2^ = 0.996) (Appendix A). *K_SV_* was calculated to be approximately 1.54 × 10^3^ M^−1^, which was comparable to other luminescent probes for the detection of nitrobenzene, such as Zn_3_(BTC)_2_ (3.957 × 10^3^ M^−1^) [35], Eu^3+^@SOF-1 (2.419 × 10^2^ M^−1^) [36], MOF 1 (3001.7 M^−1^) [37], MOF 2 (1644.1 M^−1^) [37], [(Pr_2_(TATMA)_2_·4DMF·4H_2_O]_n_ (4.1 × 10^3^ M^−1^) [38], and Ln-MOFs (1.4 × 10^3^ M^−1^) [39]. The detection limit for nitrobenzene was determined to be 1.47 × 10^−5^ M using the formula 3*σ*/*k* (*σ* = standard error; *k* = slope), which was better than most of other luminescent probes for detecting nitrobenzene, including Eu^3+^@SOF-1 (3.7 × 10^−5^ M) [36], Cd-MOF (4.2 × 10^−5^ M) [40], DF (8.12 × 10^−4^ M) [41], BDS (4.06 × 10^−4^ M) [41], and M-15 (9.91 × 10^−6^ M) [42] (Appendix A).

To better understand the fluorescence-sensing mechanism, the molecular orbitals (MOs) of LNU-23 and analytes were calculated (Figure 4). Because the energy of the lowest unoccupied molecular orbital (LUMO) of LNU-23 was higher than that of the analytes, the excited LNU-23 acted as an electron donor and the analyte served as an electron acceptor. The sustainable electron transfer from the LNU-23 to the analyte resulted in the excellent fluorescence quenching capability of LNU-23 [43]. As for B, MB, and MCB molecules, their LUMO levels were higher than those of LNU-23, implying that the electronic transitions from the excited LNU-23 to these analytes were thermodynamically forbidden, leading to poor fluorescence responses. In particular, the LUMO level of NB was the lowest among all analytes. There was a photo-induced electron transfer (PET) process of the PAF sample in the detection of NB molecules (Figure 4b), which brought about a rapid and sensitive detection of NB by LNU-23 [44,45]. In addition, LNU-23 possessed large delocalized π-conjugated units along the polymeric chain, which significantly amplified the sensing signal and improved the sensitivity of the detection of nitro compounds [27].

From the perspective of practical application, it is of great significance to prepare portable sensors that can be used for real-time and on-site detection of nitro compounds. As shown in the SEM image (Appendix A), there were many holes on the surface of the blank filter paper, which provided numerous spaces for the PAF particle loading. Through a simple immersion, LNU-23 powder was evenly loaded on a filter paper to prepare a portable indicator paper (Appendix A). As shown in Figure 5a, the indicator paper was placed on the mouth of the vial containing 10 µL of nitrobenzene. After 10 s contact, the indicator paper exhibited obvious fluorescence quenching capability under UV light (Figure 5b). The speed of detecting nitrobenzene was faster than many previously reported indicator papers, such as BDS thin film (120 s) [38], F-5 (60 s) [46], compound 1 (>25 s) [47], compound 2 (>25 s) [47], compound 9 (<60 s) [48], and DPTB-TMS (120–180 s) [2]. In addition, the LNU-23 indicator paper could be used in the detection of solid nitrobenzene. As shown in Figure 5c, the LNU-23 indicator paper was wiped with a dry fabric (5 μL NB solution dipped on a 1 × 1 cm polyester fabric). It was clearly seen that the wiped part of the indicator paper revealed an obvious fluorescence-quenching phenomenon (Figure 5d). Finally, the indicator paper performed the fluorescence detection for liquid nitrobenzene (Figure 5e). There were fluorescence changes in the indicator papers in the presence of different concentrations of nitrobenzene under UV light. With the increase in nitrobenzene concentration, the fluorescence quenching effect of the indicator paper became more obvious. These results indicated that the LNU-23-loaded indicator paper could serve as a simple, fast, and on-site to detect nitro compounds in different states of gaseous, liquid, and solid. 

## 3. Materials and Methods

### 3.1. Materials

1,3,6,8-Tetrabromopyrene (TCI), 1,2-bisphenyldiboronic acid pinacol ester (Aladdin), tetrakis(triphenylphosphine)palladium (Sigma–Aldrich), p-nitro phenol (Aladdin), p-nitrochlorobenzene (Aladdin), nitrobenzene (Aladdin), and other chemicals were purchased from commercial suppliers without further purification.

### 3.2. Synthesis of LNU-23 

A total of 300 mg of 1,3,6,8-tetrabromopyrene (0.579 mmol) and 382 mg of 1,2-bisphenyldiboronic acid pinacol ester (1.158 mmol) were poured into a 100 mL round bottom flask, and then 60 mL DMF was added into the mixture. After freezing the reaction system with liquid nitrogen, the bottle was evacuated with a vacuum pump, and then filled with nitrogen gas to normal pressure. Next, 40 mg of tetrakis(triphenylphosphine)palladium and 5 mL of K_2_CO_3_ (2 moL L^−1^) were rapidly added to the reaction system. Repeating the pumping-to-filling step three times, the reaction mixture was placed at 130 °C for 48 h, and finally cooled to room temperature. The insoluble product was successively washed with water, N,N’-dimethylformamide, tetrahydrofuran, and acetone several times to remove unreacted monomers or catalyst residues. The crude product was subjected to Soxhlet extraction with tetrahydrofuran, chloroform, and dichloromethane in turn, and then a green powder was obtained by vacuum drying, named LNU-23.

### 3.3. Preparation of LNU-23 Indicator Paper

In total, 5 mg of LNU-23 was placed in 5 mL of THF solution, and the mixture was sonicated for 10 min. The filter paper was soaked in the above solution for 12 h and dried for 12 h. The fluorescent filter paper was cut out (1 × 5 cm) to finally obtain a portable indicator paper. 

## 4. Conclusions

A fluorescent porous aromatic framework (PAFs) LNU-23 was synthesized by the Suzuki cross-coupling reaction. The highly conjugated system of LNU-23 allowed for the rapid migration of excited electrons along the polymeric chain, which amplified the weak fluorescent signal. Based on the proper LUMO of LNU-23, an efficient energy transfer from the excited LNU-23 to the analyte was achieved, leading to selective fluorescence quenching. After integrating the luminescent PAF powder into the paper, the indicator paper produced a fluorescence response to gaseous, solid, and liquid nitrobenzene. The novel fluorescence sensor has several certain characteristics of high sensitivity, high selectivity, and strong versatility, and possesses huge application potential in the outdoor detection of nitro compounds.

## Figures and Tables

**Figure 1 molecules-27-06252-f001:**
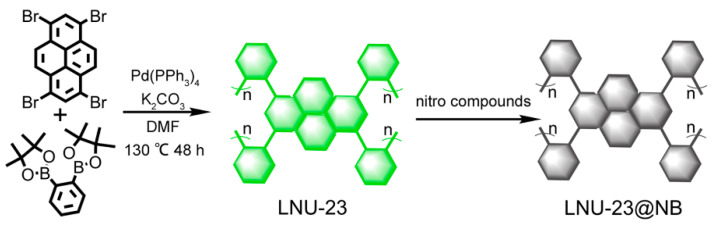
Synthesis and schematic structure of LNU-23. LNU-23 is composed of benzene rings and pyrene building blocks connected by carbon–carbon bonds. During this process, the irreversible coupling reaction tends to form defects in the architecture leading to the formation of an amorphous porous network of LNU-23.

**Figure 2 molecules-27-06252-f002:**
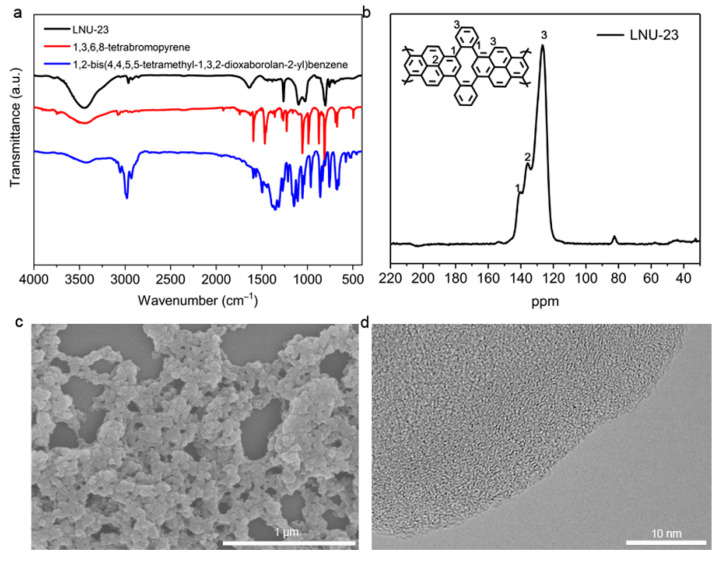
(**a**) FTIR spectra of LNU-23 and raw materials; (**b**) solid-state ^13^C NMR spectrum of LUN-23; (**c**) SEM and (**d**) TEM images of LNU-23.

**Figure 3 molecules-27-06252-f003:**
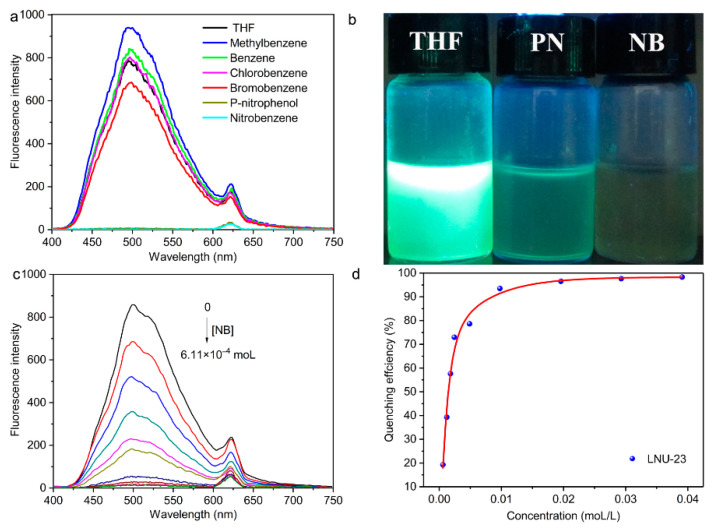
(**a**) Luminescence emission spectra of LNU-23 in the presence of various aromatic substrates. (**b**) Luminescence photographs of LNU-23 with different nitro explosives under UV irradiation at 365 nm. (**c**) Luminescence emission spectra of LNU-23 dispersion with different concentrations of nitrobenzene. (**d**) Fluorescence efficiency diagram of LNU-23.

**Figure 4 molecules-27-06252-f004:**
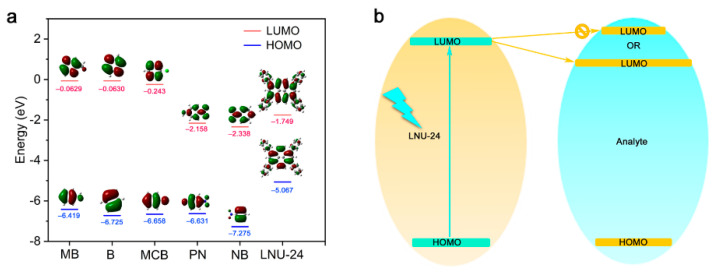
(**a**) HOMO and LUMO for tested analytes and LNU-23; (**b**) scheme of electron transfer phenomena between LNU-23 and the analyte via PET mechanism.

**Figure 5 molecules-27-06252-f005:**
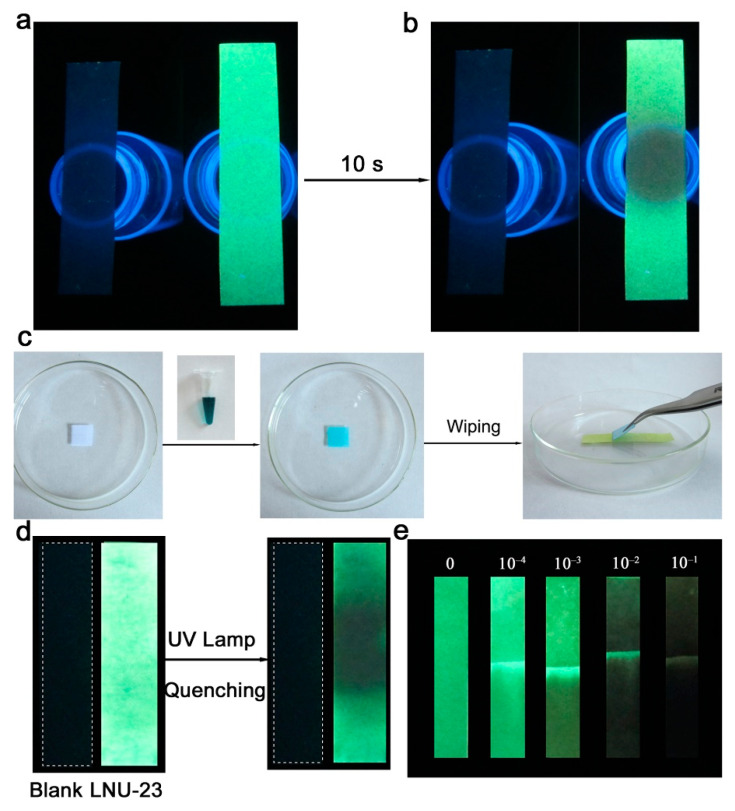
(**a**) Fluorescence diagram of indicator paper loaded with LNU-23. (**b**) Fluorescence diagram of indicator paper loaded with LNU-23 after the contact with gaseous nitrobenzene. (**c**) Process diagram of indicator paper for detecting solid nitrobenzene. (**d**) Diagram of indicator paper for detecting liquid nitrobenzene. (**e**) Fluorescence diagram of indicator papers in the presence of different concentrations of liquid nitrobenzene.

## Data Availability

All data related to this study are presented in this publication.

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
