# Peer review of "Integrating a Luminescent Porous Aromatic Framework into Indicator Papers for Facile, Rapid, and Selective Detection of Nitro Compounds"

_molecules, 2022, doi:10.3390/molecules27196252_

Round 1
Reviewer 1 Report
Comments and Suggestions for Authors
The authors report a luminescent porous aromatic framework material (LNU-23),which is synthesized by the palladium-catalyzed Suzuki cross-coupling reaction of tetrabromopyrene and 1,2-bisphenyldiborate pinacol ester. Integrated the luminescent material on the paper, then, the indicator papers reveal a fast fluorescence response to gaseous nitrobenzene within 10s, showing great potential in outdoor fluorescence detection of nitro compounds.
However, some statements are not clear and should be refracted or commented.
1 The Figure 1: Synthesis and schematic structure of LNU-23. The structure of LNU-23does not show porosity, please be supplement.
2 Figure S1. Powder X-ray diffraction pattern of LNU-23. Can you explain why there are no typical diffraction peaks
3 Figure S2. TGA plot of LNU-23 at N2 condition. When heated to 1000 degrees, LNU-23 has a 50% weight loss. Right? What is the reason? Please explain.
4 Figure S3. (a) N2 adsorption-desorption isotherm of LNU-23. The curve of gas adsorption is a little strange. Please explain what causes this adsorption behavior.
5 Figure 4. (a) HOMO and LUMO for tested analytes and LNU-23; (b) Scheme of electron transfer phenomena via PET mechanism. So this is the PET between the monomer of LNU-23 and the analyte or the PET between LNU-23 and the analyte?
Reviewer 2 Report
The authors reported an experimental work of synthesizing a luminescent porous aromatic framework material (named LNU-23). An indicator paper was prepared by integrating the luminous PAFs powder on the paper. The results show that the prepared indicator paper with rapid fluorescence response to gaseous nitrobenzene is highly welcomed for potential applications such as fluorescence detection. The material reported in this work shows great potential in the outdoor fluorescence detection of nitro compounds.
Overall, I recommend the publication of this work in the Molecules journal. Only minor revision is needed.
(1) The introduction is short. More information can be added especially on the properties, applications, ets of PAFs.
(2) The pore size is an important feature of porous aromatic framework materials. A supplementary description of the pore size distribution of the material is recommended.
(3) It is necessary to describe the range of P/P0 used for calculating BET surface area.
Reviewer 3 Report
This manuscript from Xia, Yan and coworkers reports the synthesis of a PAF material (only one example, LNU-23), its characterization through different techniques (IR, 13C_CP/MAS, XRD, BET specific surface area, absorption-emission spectra (including quantum yield of fluorescence), and its ability to act as a sensor for nitro compounds (quenching experiments, molecular calculations), including some interesting practical aspects.
This article shows a limited interest, because a single material is presented, therefore the general applicability of this type of PAF materials seems still to be determined. Despite this, the material shows good properties (it shows a good quantum yield, a good detection limit, and a fast response), although there are also some limitations (a really low BET area). The characterization seems to be correct. In principle it is potentially publishable in Molecules.
My main concern with this manuscript is that, when reading the introduction, the reader draws the conclusion that there are not previous work in the detection of nitroderivatives using PAFs (only ref 21). This is not true, and omit previous work is unfair. In a very quick search I founded the following publications, which contain explicit examples of detection of nitro compounds using PAFs: (a) Chemical Reviews 2020, 120, 8934 (Porous Aromatic Frameworks (PAFs), page 8974); (b) J. Mater. Chem A 2015, 3, 19346 (Targeted synthesis of core–shell porous aromatic frameworks for selective detection of nitro aromatic explosives via fluorescence two-dimensional response); (c) J Phys Chem Lett 2021, 12, 11050 (Biological Application of Porous Aromatic Frameworks: State of the Art and Opportunities, page 11054); (d) Chemical Journal Of Chinese Universities 2019, 40(12), 2456 (Facile Strategy to Prepare Fluorescent Porous Aromatic Frameworks for Sensitive Detection of Nitroaromatic Explosives; this last paper is from the same authors). Therefore, the authors have to do a critical analysis about the relevance of the work here presented with respect to the previous work. The mention of these papers is not enough, a true comparison needs to be done.
Moreover, the experimental description (apparatus and methods) of the measurement of the quantum yield, and details about the type and level of molecular calculations performed to determine HOMO and LUMO, must be included (at least in the Supporting Information!).
Round 2
Reviewer 3 Report
The authors have improved the manuscript and now it is publishable as is.